# Microwave-Assisted Catalytic Method for a Green Synthesis of Amides Directly from Amines and Carboxylic Acids

**DOI:** 10.3390/molecules25081761

**Published:** 2020-04-11

**Authors:** Adam P. Zarecki, Jacek L. Kolanowski, Wojciech T. Markiewicz

**Affiliations:** Institute of Bioorganic Chemistry, Polish Academy of Sciences, Noskowskiego 12/14, 61-704 Poznań, Poland; azarecki@man.poznan.pl

**Keywords:** amide synthesis, direct amidation, microwave, green chemistry, solvent free

## Abstract

Amide bonds are among the most interesting and abundant molecules of life and products of the chemical pharmaceutical industry. In this work, we describe a method of the direct synthesis of amides from carboxylic acids and amines under solvent-free conditions using minute quantities of ceric ammonium nitrate (CAN) as a catalyst. The reactions are carried out in an open microwave reactor and allow the corresponding amides to be obtained in a fast and effective manner when compared to other procedures of the direct synthesis of amides from acids and amines reported so far in the literature. The amide product isolation procedure is simple, environmentally friendly, and is performed with no need for chromatographic purification of secondary amides due to high yields. In this report, primary amines were used in most examples. However, the developed procedure seems to be applicable for secondary amines as well. The methodology produces a limited amount of wastes, and a catalyst can be easily separated. This highly efficient, robust, rapid, solvent-free, and additional reagent-free method provides a major advancement in the development of an ideal green protocol for amide bond formation.

## 1. Introduction

Amide bonds are among the most widely abundant and fascinating types of linkages in organic synthesis and nature [1,2,3,4]. They constitute the backbone of peptides and proteins and are important elementary linkages in many natural products and polymers. In addition, thanks to their stability in biological environments, they are often used in the construction of various drugs, insecticides, nutraceutics, and chemical tools to study and modify biology [5,6,7,8].

Nature uses enzymes to form amides from carboxylic acids and amines in a catalytic reaction. For example, the translation of mRNA to proteins in ribosomes, which is one of the key processes in biology, is catalyzed by peptidic transferase, which extends the peptide chain by linking a new amino-acid in the form of aminoacyl-tRNA with an N-terminus of the peptide through an amide bond [9,10].

To date, many different methods have been developed to enable the formation of various amide linkages in synthetic laboratories. In particular, arguably the biggest drive to investigate these methods came from the success of solid-phase peptide synthesis (SPPS) [11]. Nevertheless, most of them require an active form of carboxylic acid (formed either in a separate synthetic step or in situ, for example, in a reaction with a so-called “coupling reagent”), which is subsequently attacked by a nucleophile in the next step [12,13]. Moreover, common coupling reagents (e.g., carbodiimides, phosphonium, or uronium salts [14,15,16]) are often rather expensive, toxic, and are used in stoichiometric quantities or in excess, leading to a poor atom economy. This, in combination with the need for purification from various by-products, makes these methods challenging and costly, pushing many scientific groups to investigate faster, more straightforward, and therefore “greener” routes for the formation of amide bonds.

Green chemistry focuses on designing products and processes to minimize the environmental and economic impact. This can be achieved by a) avoiding difficult to obtain and hazardous starting materials and additional reactants, b) increasing the efficiency of reactions and minimizing by-product formation (e.g., the use of selective catalysts), c) reducing energy consumption, and d) minimizing hazardous waste (e.g., using safer or even no solvent and ensuring facile isolation). In this context, an ideal amide bond formation reaction would be:Directly between carboxylic acids and amines and would not require the isolation of intermediates;Catalytic and atom efficient (e.g., no coupling reagents);Waste free (e.g., no solvents and minimal purification); andFast (no extended reaction times) [17,18].

Due to the ecological and economic advantage of obtaining amides directly from carboxylic acids and amines, several groups have explored this possibility in recent years. The resulting methods include, for example, reflux in toluene in the presence of a Zr(IV) catalyst [19]. A similar transformation was achieved at a lower temperature (reflux of diethyl ether/toluene) in the presence of molecular sieves, but it also required an expensive catalyst (Hf(IV) or Zr(IV)) and long reaction times measured in days [20].

Microwave-assisted rapid organic reactions are an attractive green technology with great potential to meet many of the above-mentioned criteria [21]. It has been noticed that microwave synthesis increases the yield of the reaction, reduces the formation of wasteful by-products, increases the reproducibility of the reaction, and reduces reaction times. It allows also for the use of solid substrates (provided an adequate melting point of at least one reagent) without the requirement of their previous dissolution [22,23,24]. Consequently, microwave radiation was recently used for the synthesis of primary amides from carboxylic acids and urea in the presence of catalytic amounts of cerium ammonium nitrate [25,26], as well as for direct amidation, allowing for shortening of the reaction times and more energy-efficient heating [27,28]. However, these methods require a solvent and the presence of near-stoichiometric amounts of TiO_2_ (33.3 wt%) or excess silica gel, which facilitate the transformation, contributing to significant waste generation. In addition, in order to isolate products, column chromatography was needed.

## 2. Results and Discussion

In this paper, we report a microwave-assisted synthetic protocol for the preparation of amides directly from carboxylic acids and amines in very good to excellent yields at reaction times significantly shorter (one to few hours) than those of many previously reported direct amidations. Our protocol also eliminates the need for a solvent or any additional reagents, a significant advantage over previously reported methods. In addition, the reaction occurs in the presence of trace amounts (0.1–2 mol%) of ceric ammonium nitrate (CAN), a common catalyst for microwave reactions [29,30], or even without a catalyst at all (for some substrates) (Scheme 1). Pure products were isolated through extraction with ethyl acetate/water mixtures (acidic, basic, or saline) without the need for tedious column chromatography. Additionally, this process separated any traces of the catalyst, enabling the recovery of cerium, and minimizing any potential environmental threat and increasing the economics of the process. As such, this methodology is a major leap forward in the endeavor of developing an ideal green synthesis of amide bonds.

This robust reaction was investigated in more detail to evaluate the role of the environmental conditions and chemical components of the process. The aim of these efforts was to maximize the reaction efficiency, while minimizing the time and energy needed for the reaction to occur, making this transformation as “green” as possible. We selected *p*-toluidine and an appropriate acid as model substrates and varied the concentration of the catalyst and reaction conditions. A summary of the results of these experiments is provided in Table 1, Table 2 and Table 3. The optimal conditions of the reactions for each acid substrate are marked in red.

Initially, for phloretic acid, reactions were performed with ceric ammonium nitrate (2 mol%) as the catalyst and without any solvent, with temperatures varying from 60–65 °C to 160–165 °C in 20-°C intervals (Table 1, entries 1–5). As shown in Table 1, an improvement in the yield of the reaction (assuming the same reaction times and other conditions being constant) was observed with an increasing temperature until 120–125 °C, above which no further difference could be noticed. Next, a decrease in reaction times was investigated (Table 1, entry 4 and 6), showing that 2 h is the optimal time for this particular reaction. We also tested the influence of the amount of catalyst on the yield of the reaction (Table 1, entry 7–11) and showed, remarkably, that even in its absence, the reaction occurs in very good yields, which can be further improved by increasing the reaction times. In fact, the same yield can be obtained with 0.1 mol% of CAN and 5 h of reaction as with 2 mol% CAN in 2 h. Therefore, it is now up to the chemist to decide whether, for his or her needs, it is better to continue a reaction for longer but avoid the use of a catalyst, or reduce the reaction time by half or more by adding a catalyst.

Similar behavior was observed for another alkyl-carboxylic acid, phenylacetic acid (Table 2). The reaction was shown to yield the desired product without the use of a catalyst within 2 h, and an increase of the temperature from 120–125 °C to 160–165 °C further improved the yield. However, in order to ensure near-quantitative transformation of the substrates to the products, the reaction had to be performed in the presence of 2 mol% of a catalyst for 2 h and in 160–165 °C (higher than in the case of phloretic acid). 

We also used an aryl-carboxylic acid, benzoic acid, and performed a similar screen of the conditions (Table 3). We found that in order to obtain near-quantitative yields similar to those with phloretic acid, 2 mol% of catalyst and a temperature of 160–165 °C (as in the case of phenylacetic acid) was required, together with a prolonged (5 h) reaction time. Interestingly, for this substrate, the lack of a catalyst (or trace amounts of it) inhibited the reaction (unlike in the case of the alkyl-carboxylic acids described above). This indicates that reactions with benzoic acid derivatives occur significantly slower, but the conditions can be adapted to obtain higher yields. To further investigate the role of CAN in this reaction, we monitored its progress at different time points with NMR (the reaction between benzoic acid and benzylamine was chosen as a model; Appendix A). Initially, no reaction occurs if catalyst is not present and the reaction mixture does not heat up in a microwave (i.e., does not reach the desired temperature). In the presence of CAN, signals from the desired product (amide) start to appear already after 30 min from the beginning of the reaction (according to the NMR; Appendix A), reaching the maximum in 2 h. Further heating does not lead to any change. This confirms the important role of CAN in the reaction of an amine with benzoic acid under our experimental conditions, as also reported previously for other amide bond formation protocols. Nevertheless, to date, the exact mechanism of its involvement remains the subject of debate. One possible hypothesis could point to the catalytic mechanism in which cerium(IV) ion in CAN acts as a Lewis acid or indirectly leads to the formation of a Brønsted acid [31]. It is also possible to postulate an additional role of CAN as an acidic catalyst through the “liberation” of nitric acid, acting analogously to carboxylic acid in a mechanism proposed by Charville et al. [32]. In either case, acidic catalysis, whether involving the activation of carbonyl carbon in carboxylic acid and/or the formation of a complex acting as a better leaving group, seems to be the most plausible explanation. Nevertheless, deeper analysis of this mechanism would require significant experimental and theoretical effort, which is outside the scope of this manuscript, but could constitute a separate publication in the future.

To further validate the protocol, we selected a range of aromatic (anilines) and aliphatic (benzylamines and alkylamines) amines, which differ in their nucleophilicity (with phenyl rings bearing H, Me, OMe, or F substituents). Benzoic acid and phloretic acid were used as models of aromatic and aliphatic carboxylic acids, respectively. The latter also bears a phenolic group, demonstrating the compatibility of the protocol with such functionalities. In addition, phloretic acid has been reported, together with its corresponding primary amide, as a new promising bioactive scaffold, but the number of its derivatives remains limited [33].

The results of these experiments are summarized in Table 4. In order to enable a direct comparison of the yields of these reactions between different acid substrates, we decided to select identical intermediate reaction conditions for all experiments, which were as follows: 160–165 °C, 2 h, 2 mmol (1 equivalent (eq) of carboxylic acid and 2.1 eq of amine, 2 mol% of a catalyst. In the case of the reactions of benzylamine and *p*-methylbenzylamine with phloretic acid as well as putrescine (diamine) with both acids, it was found that the use of excess acid was preferred to ensure full conversion of the amine (Appendix A). This allowed for an extraction-based purification, which otherwise (in case of standard protocol) required extra column chromatography for the separation of the mono (benzylamines) and diamide (putrescine) products.

Analysis of the reaction yields with respect to the chemical character of the substrates points to some interesting trends. The highest yields (above 90%) were observed for the reactions between phloretic acid and anilines, with little effect of the electron-withdrawing/electron-donating character of the analogous amine (the somewhat higher yields for *p*-methoxyaniline over *p*-fluoroaniline are not significant enough for reliable conclusions). The yields were similar for the phloretic acid reactions with alkylamines and benzylic amines bearing electron-withdrawing groups, while other benzylic amines gave lower yields. The tendency for higher yields with benzylamines bearing electron-poor phenyl rings over electron-rich analogues, observed for phloretic acid, translates also to reactions with benzoic acid. In general, reactions with benzoic acid gave significantly lower (although still satisfactory) yields than in the case of phloretic acid. The higher yields of this reaction for alkyl over aromatic carboxylic acids were confirmed by additional experiments with 4-methylpentanoic acid and phenylacetic acid, each with one aniline (anisidine) and one benzylamine (*p*-fluorobenzylamine)-type amines (Table 4, bottom entries). Indeed, both acid substrates gave very high yields of the reactions, similar to the ones observed for phloretic acid.

While a variation in the yields of the reactions seems to stem from the electronic properties of the carboxylic acid substrates (e.g., preference for aliphatic vs. aromatic), other factors (e.g., solubility, conformational flexibility of the substrates, polarity, CAN-complexation capacity of amine substrates, and others) cannot be excluded. Importantly, we observed that in our conditions, the reaction can take place even between two substrates, which are solids at room temperature, provided that the melting point of at least one of them is below the temperature of the reaction. However, in such cases, the yields seem to drop. In particular, the yields of the reactions of benzoic and phenylacetic acids with anisidine (all substrates are solids) are relatively lower than for other reactions of these acids but with amines, which are liquid at room temperature (approximately 50% vs. 75% and 85% vs. >90%, respectively). In contrast, the reaction of solid anisidine with 4-methylpentanoic acid, which is a liquid at (room temperature (rt), led to the isolation of a product in a very high yield (98%).

We also confirmed in preliminary experiments that the protocol developed within this work can be applied to secondary amines. However, the optimized conditions of the reaction, together with the purification protocols, require further elaboration, due to the lower polarity of tertiary amides in comparison to secondary ones. This work is in progress and will be reported in the future in a separate manuscript.

In order to further extend the utility of our protocol, while maintaining “green” reaction conditions, we investigated the reactivity in the presence of water as an environmentally friendly solvent. In solvent-free conditions, the minimal amount of the substrates required for a reaction to occur was approximately 200 mg (for lower quantities, the reaction mixture could not be efficiently heated up in a microwave reactor). The addition of water allowed us to perform reactions efficiently down to the scale of a few dozens of milligrams, and also enabled amide bond formation between substrates with higher melting points (Appendix A). The yields of these reactions were comparable to the yields obtained with other similar substrates. 

The biological activity of all prepared amide derivatives is under investigation in 2D cultures of human cells as well as induced pluripotent (iPS) cells as biological models, and the results of these studies, combining a larger library of amide derivatives, will be published elsewhere.

## 3. Materials and Methods 

### 3.1. Chemistry—General Information

All chemicals were purchased from Sigma-Aldrich (St. Louis, MO, USA) and Eurisotop Saint-Aubin, France) (NMR solvents) and used without further purification. ^1^H NMR and ^13^C NMR spectra were recorded on an AVANCE III Bruker spectrometer (Bruker Corporation, Billerica, Massachusetts, USA) in DMSO-d*_6_* at frequencies of 500 and 126 MHz, respectively. Spectra were processed using MestReNova Version 9.0.1-13254. NMR data were reported as follows: Chemical shift (δ), multiplicity (recorded as br, broad; s, singlet; d, doublet; t, triplet; q, quadruplet, and m, multiplet), coupling constants (*J* in Hertz, Hz), and integration. The chemical shifts are expressed in parts per million (ppm) and reported in relation to a residual solvent peak (2.50 and 39.5 ppm for ^1^H and ^13^C NMR in DMSO-*d_6_*, respectively). High-resolution mass spectrometry (HRMS) was performed on a Q-Exactive Orbitrap mass spectrometer (Thermo Fisher Scientific, Bremen, Germany) equipped with a TriVersa NanoMate robotic nanoflow ESI ion source (Advion BioSciences ltd., Ithaca, NY, USA). The nanoelectrospray chips with a nozzle diameter of 5.5 µm were used in order to obtain a stable ion spray and Chipsoft software (ver. 8.3.1.1018) was applied to control the ESI ion source. Microwave experiments were performed in a commercially available open microwave reactor (Magnum II) from ERTEC-Poland Dr. Edward Reszke, Wroclaw, Poland. Microwave power was smoothly regulated in the range from 0 to 750 W at a field frequency of 2.45 GHz. The temperature was monitored by a pyrometer and by monitoring the vapor temperature using a K-type thermocouple. Reactions were carried out at atmospheric pressure in regular glass flasks equipped with a reflux condenser glass topped with a free airflow tube with a drying agent.

### 3.2. Synthesis of Amides—General Procedure

Amine (4.2 mmol), carboxylic acid (2 mmol), and catalyst (ceric ammonium nitrate, CAN) (2 mol%) were added to an empty flask equipped with a reflux condenser at atmospheric pressure and placed in a microwave set to maintain a constant temperature in the range of 160–165 °C for a given time period (microwave power set up to 480 W but was smoothly and automatically controlled by the software to keep the target temperature constant). After 2 h, the reaction mixture was allowed to cool to room temperature, and subsequently, 25 mL of ethyl acetate were added. The organic phase was washed with 3 × 15 mL of 2 M aqueous HCl, 3 × 15 mL of saturated aqueous NaHCO_3_, and 3 × 15 mL of saturated aqueous NaCl; filtered; and solvent from the organic fraction was removed under reduced pressure to obtain the pure product. The compounds were fully characterized by ^1^H NMR, ^13^C NMR, and high-resolution mass spectrometry (HRMS) (data shown in Appendix A—characterization of products (amides)—compounds 1–24).

As examples, the NMR spectra and HRMS results for two novel compounds that were obtained within the scope of this work are shown below. A complete set of data for all new compounds not reported previously in the literature (2, 4, 6, 8, 12, 14, 16, 18, 20, 22, and 24) and other synthesized compounds is provided in the Appendix A. 

*3-(4-hydroxyphenyl)-N-phenylpropanamide (2):* Compound 2 was obtained in the form of a solid, following the general procedure described above (yield = 94%, 92% and 96%—3 repeats). ^1^H NMR (500 MHz, DMSO-*d**_6_*) δ = 9.86 (s, 1H), 9.15 (s, 1H), 7.62–7.54 (m, 2H), 7.28 (*t*, *J* = 7.9 Hz, 2H), 7.06–6.99 (m, 3H), 6.70–6.62 (m, 2H), 2.80 (t, *J* = 7.7 Hz, 2H), 2.56 (t, *J* = 7.7 Hz, 2H). ^13^C NMR (126 MHz, DMSO-d*_6_*) δ = 171.0, 155.9, 139.7, 131.7, 129.5, 129.1, 123.4, 119.5, 115.5, 38.9, 30.6. HRMS [M + H]^+^ calcd for C_15_H_15_NO_2_: 242.1175, found: 242.1154.

*3-(4-hydroxyphenyl)-N-(p-tolyl)propanamide (4)*: Compound 4 was obtained in the form of a solid, following the general procedure described above (yield = 89 and 95%—2 repeats). ^1^H NMR (500 MHz; DMSO-*d**_6_*) δ = 9.77 (s, 1H), 9.15 (s, 1H), 7.46 (d, *J* = 8.2 Hz, 2H), 7.08 (d, *J* = 8.2 Hz, 2H), 7.03 (d, *J* = 8.2 Hz, 2H), 6.67 (d, *J* = 8.4 Hz, 2H), 2.79 (t, *J* = 8.7, 6.7 Hz, 2H), 2.53 (t, *J* = 8.7, 6.8 Hz, 2H), 2.23 (s, 3H). ^13^C NMR (126 MHz, DMSO-d*_6_*) δ = 170.8, 155.9, 137.2, 132.3, 131.7, 129.5, 129.5, 119.6, 115.5, 38.9, 30.6, 20.9. HRMS [M + H]^+^ calcd for C_16_H_17_NO_2_: 256.1321, found: 256.1332.

## 4. Conclusions

In conclusion, we developed a cheap, universal, high-yielding, and green method for the direct amidation of carboxylic acids with different amines. The use of microwave energy remarkably facilitates this otherwise very difficult direct transformation, without the need for any coupling reagents. Its versatility in terms of the substrates extends from electron rich to electron poor, aliphatic and aromatic amines and carboxylic acids. This, together with the lack of solvents (except water in some cases) and even catalyst, fast reaction kinetics (significantly faster than under standard conditions where direct amidations with catalyst last for days [18,19]), simplicity of purification, and scalability (from dozens of milligrams to grams), combined with the high atom efficiency puts the reported protocol at the forefront of synthetic methodologies for the sustainable formation of amide bonds.

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
