# Peer review of "Microwave-Assisted Catalytic Method for a Green Synthesis of Amides Directly from Amines and Carboxylic Acids"

_molecules, 2020, doi:10.3390/molecules25081761_

Round 1
Reviewer 1 Report
the article is of interest, however it must be modified for acceptance:
Include italics in concepts such as J, d6, p ...
Include NMR and characterization data in the text for new compounds
Include in results and discussion spectroscopic and spectrometric analysis of at least one compound as a generic example
enhance the importance of the proposed methodology compared to that already existing at the level of reaction time, performance, purity, etc.
Include potential biological activity of these compounds previously reported or under study
Author Response
The authors appreciate and sincerely thank the referee for valuable time and reviewof our manuscript. Based on the comments, we revised the manuscript providing additional explanations of some of the aspects of the work for better clarity and understanding, as suggested by the reviewer.
Point to point answers start below. In this response the comments from referees are written in italic, and our answers follow. The changes within the revised manuscript will be shown in blue and underlined.
Comment. Include italics in concepts such as J, d6, p ...
Our reply. We introduced suggested modification throughout the manuscript and the supplementary part.
Comment. Include NMR and characterization data in the text for new compounds. Include in results and discussion spectroscopic and spectrometric analysis of at least one compound as a generic example.
Our reply. All synthesized compounds are characterized by NMR and both the data and spectra are included in the supplementary part, together with an information whether the compounds are new or were reported earlier. Following reviewer’s request, data for two new amides were included into the manuscript as example.
Comment. Enhance the importance of the proposed methodology compared to that already existing at the level of reaction time, performance, purity, etc.
Our reply. The appropriate statements underlying the advantages of the proposed methodology were included into the revised manuscript. In comparison to previous literature in the topic our protocol combines multiple advantages, including (i) direct and rapid amidation from amines and carboxylic acids (refs 1 and 2, below) (ii) lack or mimimal amounts of catalysts (refs 3-6, below), (iii) no requirement of the solvent or other additional reagents like SiO2 or TiO2 minimising waste production (refs 3-6, below).
- Reddy S.; Raghu M.; Nagaraj A. Ceric ammonium nitrate (CAN) promoted efficient solid phase synthesis of amide derivatives: A green approach. Indian Journal of Chemistry 2008, 47B, 315-318.
- Gupta N.; Singh M. Microwave Assisted High Speed Chemistry, A New Technology for Pharmaceutical Industry. Asian J. Chem. 2012, 24(12), 5937-5938.
- Lundberg H.; Adolfsson H. Hafnium-Catalyzed Direct Amide Formation at Room Temperature. ACS Catal. 2015, 5, 3271-3277.
- Allen C. L.; Chhatwal A. R.; Williams J. M. J. Direct amide formation from unactivated carboxylic acids and amines. Commun. 2012, 48, 666–668.
- Ojeda-Porras A.; Hernández-Santana A.; Gamba-Sánchez D. Direct amidation of carboxylic acids with amines under microwave irradiation using silica gel as a solid support. Green Chem. 2015, 17,3157.
- Gaudino E. C.; Carnaroglio D.; Nunes M. A. G.; Schmidt L.; Flores E. M. M.; Deiana C.; Sakhno Y.; Martrac G.; Cravotto G. Fast TiO2-catalyzed direct amidation of neat carboxylic acids under mild dielectric heating. Sci. Technol. 2014, 4, 1395.
Comment. Include potential biological activity of these compounds previously reported or under study.
Our reply. We thank very much for this comment. Indeed, biological activity of these compounds are also of our interest, and as such they are currently under an investigation as a part of a larger library of amide derivatives in cellular assays including cancer cell lines and IPs and as such will be published separately as a part of a larger study. Here, we intentionally focused on the methodology itself.

Reviewer 2 Report
In the manuscript, the Authors have investigated on a new method of the direct synthesis of amides from carboxylic acids and amines under solvent-free conditions using small amount of ceric ammonium nitrate (CAN) as a catalyst.
The work is properly done, clearly laid out and in logical sequence. The paper is written in a good language and style, laying the basis for future work in the field and I am sure that the approach used provides an useful contribute to the Molecules’s readers.
I have only one criticism about the references which in my opinion are not updated.
Nevertheless, I would recommend the paper for publication in the journal.
Author Response
The authors appreciate and sincerely thank the referee for valuable time and review of our manuscript. Based on the comments, we revised the manuscript providing additional explanations of some of the aspects of the work for better clarity and understanding, as suggested by the reviewer.
Point to point answers start below. In this response the comments from referees are written in italic, and our answers follow. The changes within the revised manuscript will be shown in blue and underlined.
Comment. I have only one criticism about the references which in my opinion are not updated.
Our reply. Authors appreciate very much the reviewer’s comment concerning the literature search and have added some new references to the list of references in the revised version. However, despite our efforts we could not find newer reviews that cover contemporary achievements in the microwave assisted and CAN catalyzed syntheses besides the review published in 2010 - Sridharan, V.; Menéndez, J.C. Cerium(IV) Ammonium Nitrate as a Catalyst in Organic Synthesis. Chem. Rev. 2010, 110, 3805–3849; doi 10.1021/cr100004p.
There are practically no reports, at least we are not aware of, addressing the question of the reaction mechanism of CAN catalyzed reactions similar to another paper (Charville, H.; Jackson, D.A.; Hodges, G.; Whiting, A.; Wilson, M.R. The Uncatalyzed Direct Amide Formation Reaction - Mechanism Studies and the Key Role of Carboxylic Acid H-Bonding. European J. Org. Chem., 2011, 30, 5981-5990; doi 10.1002/ejoc.201100714) again published 10 years ago.
A similar question was addressed by Thomasso Marcelli (Marcelli, T. Mechanistic insights into direct amide bond formation catalyzed by boronic acids: halogens as Lewis bases. Angew. Chem. Int. Ed. Engl. 2010, 49, 6840–3; doi 10.1002/anie.201003188), again in 2010.

Reviewer 3 Report
This paper describes a useful high-yielding method of formation of amides that should prove useful in synthetic labs (provided they have a microwave reactor). The experimental design is good and deductions pertaining to reaction conditions are sound as are conclusions re aromatic v aliphatic acid as well as electronic effects of amine substituents. With respect to Table 2, I think the importance of presence of the catalyst is not compelling as the improvement in yield is minimal at each temperature.
I think the paper would be improved somewhat if there were some discussion as to the mechanism of the reaction and the role of Ce(IV) in the process. In addition, what might be the nature of and the role of the amine-acid-Ce(IV) complex that is evident in the mass spectra.
The ms reaction studies detailed in S2 of the Supplementary material seem unconvincing and perhaps superfluous other than detecting the presence of the Ce(IV) complex. There is no ion corresponding to the product N-benzylbenzamide (m/z 211) and no obvious changes in intensity, unlike the NMR spectra. I note that the labelling of benzylamine methylene in some of the NMR spectra seems odd unless concentration effects are playing a role.
NMR spectra of the products are well presented apart from a number of solvent peaks.
The manuscript is fairly well written apart from a few incorrectly hyphenated words (sol-vent, per-formed) and spelling errors (spectrommetry)
Author Response
The authors appreciate and sincerely thank the referee for valuable time and review of our manuscript. Based on the comments, we revised the manuscript providing additional explanations of some of the aspects of the work for better clarity and understanding, as suggested by the reviewer.
Point to point answers start below. In this response the comments from referees are written in italic, and our answers follow. The changes within the revised manuscript will be shown in blue and underlined.
Comment. With respect to Table 2, I think the importance of presence of the catalyst is not compelling as the improvement in yield is minimal at each temperature.
Our reply. Authors appreciate very much all the reviewer’s comments. We agree with the conclusion that the effect of CAN is hardly pronounced or visible at all when the yields reported in Table 2 entries 1&2 vs. 3&4 are addressed. Yet, the similar inspection of Table 3 entries 5-8 stronger supports the conclusion that CAN addition is crucial. Therefore, in order to compare the results of amide formation with different substrates, we decided to keep the conditions similar, even if CAN, for some substrates, was found not necessary. In fact, we have also mentioned it in the manuscript, underlying, that “it is now to the chemist to decide” whether for some reactions, they prefer to use catalyst to gain some time / yield or prefer to perform a reaction without a catalyst.
Comment. I think the paper would be improved somewhat if there were some discussion as to the mechanism of the reaction and the role of Ce(IV) in the process. In addition, what might be the nature of and the role of the amine-acid-Ce(IV) complex that is evident in the mass spectra.
Our reply.
Thank you very much for your comment.In the present version, some aspects of the reaction mechanism were mentioned e.g. by referring to reaction mechanisms of direct reaction carboxylic acids and amines discussed by Charville et al., and by Thomasso Marcelli.
- Charville, H.; Jackson, D.A.; Hodges, G.; Whiting, A.; Wilson, M.R. The Uncatalyzed Direct Amide Formation Reaction - Mechanism Studies and the Key Role of Carboxylic Acid H-Bonding. European J. Org. Chem., 2011, 30, 5981-5990;
doi 10.1002/ejoc.201100714 - Marcelli, T. Mechanistic insights into direct amide bond formation catalyzed by boronic acids: halogens as Lewis bases. Chem. Int. Ed. Engl. 2010, 49, 6840–3;
doi 10.1002/anie.201003188

Reviewer 4 Report
This manuscript described the microwave-assisted direct synthesis of amides from carboxylic acids and primary amines under solvent-free conditions using ceric ammonium nitrate (2 mol%) as a catalyst in good to excellent yields.
The manuscript was written and presented in a good structure. However, the main idea of this work was published previously in the literature. Two published articles described the same route of the direct amidation reaction using a series of carboxylic acid derivates and urea as a source of primary amines (see the attached files). Also, the same synthesis procedure, including the amount of CAN (0.2 mol%) has been used as a key role in this coupling reaction, as recommended in the previously published articles.
In Page 2, Lines 892-84, the author mentioned the following sentence (In addition, the reaction occurs in the presence of trace amounts (0 – 2 mol%) of ceric ammonium nitrate (CAN) – a common catalyst for microwave reactions [26-29] or even without a catalyst at all (for some substrates) (Scheme 1).
In fact, Reference 26 described the same work in this manuscript. Unfortunately, the authors did not mention that CAN has been used previously as a potential catalyst for the amidation reaction under microwave irradiation.
Here are the articles from the literature:-
- Gupta, N., Singh, M. Asian Journal of Chemistry; Vol. 24, No. 12 (2012), 5937-5938.
- Reddy C. S.; Raghu M.; Nagaraj A. Ceric ammonium nitrate (CAN) promoted efficient solid phase synthesis of amide derivatives: A green approach. Indian Journal of Chemistry 2008, 47B, 315-318.
Therefore, there is no novelty in this manuscript. In current form I would reject this manuscript to publish in molecules.

Author Response
The authors appreciate and sincerely thank the referee for valuable time and review of our manuscript. Based on the comments, we revised the manuscript providing additional explanations of some of the aspects of the work for better clarity and understanding, as suggested by the reviewer.
Point to point answers start below. In this response the comments from referees are written in italic, and our answers follow. The changes within the revised manuscript will be shown in blue and underlined.
Comment. The manuscript was written and presented in a good structure. However, the main idea of this work was published previously in the literature. Two published articles described the same route of the direct amidation reaction using a series of carboxylic acid derivates and urea as a source of primary amines (see the attached files). Also, the same synthesis procedure, including the amount of CAN (0.2 mol%) has been used as a key role in this coupling reaction, as recommended in the previously published articles.
In fact, Reference 26 described the same work in this manuscript. Unfortunately, the authors did not mention that CAN has been used previously as a potential catalyst for the amidation reaction under microwave irradiation.
Here are the articles from the literature:-
- Gupta, N., Singh, M. Asian Journal of Chemistry; Vol. 24, No. 12 (2012), 5937-5938.
- Reddy C. S.; Raghu M.; Nagaraj A. Ceric ammonium nitrate (CAN) promoted efficient solid phase synthesis of amide derivatives: A green approach. Indian Journal of Chemistry 2008, 47B, 315-318.
Therefore, there is no novelty in this manuscript. In current form I would reject this manuscript to publish in molecules.
Our reply. The main point of the reviewer’s criticism relies on referring to paper No 2 (Reddy et al.) i.e. reference 26 within our manuscript. We agree that this should be expressed in a clear manner that paper No 2 describes the synthesis of amides from carboxylic acids using CAN as a catalyst. This was now introduced to the revised version of the manuscript together with adding another reference brought to the discussion by the reviewer (No 1, Gupta and Singh).
However, we believe, this criticism of lack of sufficient novelty is not justified and we hope to convince the Reviewer and the Academic Editor of Molecules journal that this would be a conclusion too far-reaching.
The two papers quoted above describe the synthesis of amides from carboxylic acids and UREA (not direct amidation of amines reported here in our manuscript).
None of amides described in these two publications were SUBSTITUTED amides. All are of general structure RCONH2 i.e. PRIMARY amides. The reviewer writes “urea as a source of primary amines”…However, urea can be regarded as source of AMMONIA, but not AMINES. If one would quote precisely Reddy et al. write “synthesis of amides from urea, a source of ammonia, and carboxylic acids.”(page 315, right column, line 18 from the bottom), significantly limiting a scope of products which could be synthesized according to the protocol (only to primary amides).
Therefore, we disagree with the judgment denying novelty of our approach involving the use of amines as a substrate, particularly as argumentation provided in the review misreferred previous work, bringing it as an unjustified equivalence. Of course, one could profit from the paper of Reddy et al. differently, however, others but us did not.
There are many methods described in organic chemistry that are similar and very similar, but similar does not mean equal. One of substrates is different. One of products is similar. UREA IS NOT AN AMINE rather it is an amide. And it is PRIMARY amide strongly limiting the scope of accessible materials. All products described in our manuscript are SECONDARY amides and even TERTIARY amides (both of which, in general, have significantly higher industrial and biological utility) but not primary ones. This is simply another story or at least clearly a different one.
We hope that the Reviewer will share this point of view and rather stick to what he has written in the first sentence of the second paragraph of his/her review: “The manuscript was written and presented in a good structure.” and withdraw his denial of publication of this new work in Molecules journal.

Round 2
Reviewer 4 Report
The authors added the previously published work in a clear sentence to the manuscript. Therefore, we could accept this work to publish in molecules.